# A Multi-Level Strategy Based on Metabolic and Molecular Genetic Approaches for the Characterization of Different *Coptis* Medicines Using HPLC-UV and RAD-seq Techniques

**DOI:** 10.3390/molecules23123090

**Published:** 2018-11-27

**Authors:** Furong Zhong, Chan Shen, Luming Qi, Yuntong Ma

**Affiliations:** 1State Key Laboratory Breeding Base of Systematic Research, Development and Utilization of Chinese Medicine Resources, Chengdu University of Traditional Chinese Medicine, Chengdu 611137, China; zhongfurong005@163.com (F.Z.); 18408257881@163.com (C.S.); 18669326801@163.com (L.Q.); 2School of Pharmacy, Chengdu University of Traditional Chinese Medicine, Chengdu 611137, China; 3The engineering and technology research center for the protection and development of Yalian resources in Sichuan Province, Meishan 620360, Sichuan, China

**Keywords:** *Coptis* plants, HPLC-UV, RAD-seq, multivariable statistical method

## Abstract

*Coptis* plants (Ranunculaceae) to have played an important role in the prevention and treatment human diseases in Chinese history. In this study, a multi-level strategy based on metabolic and molecular genetic methods was performed for the characterization of four *Coptis* herbs (*C. chinensis*, *C. deltoidea*, *C. omeiensis* and *C. teeta*) using high performance liquid chromatography-ultraviolet (HPLC-UV) and restriction site-associated DNA sequencing (RAD-seq) techniques. Protoberberine alkaloids including berberine, palmatine, coptisine, epiberberine, columbamine, jatrorrhizine, magnoflorine and groenlandicine in rhizomes were identified and determined based on the HPLC-UV method. Among them, berberine was demonstrated as the most abundant compound in these plants. RAD-seq was applied to discover single nucleotide polymorphisms (SNPs) data. A total of 44,747,016 reads were generated and 2,443,407 SNPs were identified in regarding to four plants. Additionally, with respect to complicated metabolic and SNP data, multivariable statistical methods of principal component analysis (PCA) and hierarchical cluster analysis (HCA) were successively applied to interpret the structure characteristics. The metabolic variation and genetic relationship among different *Coptis* plants were successfully illustrated based on data visualization. Summarily, this comprehensive strategy has been proven as a reliable and effective approach to characterize *Coptis* plants, which can provide additional information for their quality assessment.

## 1. Introduction

Traditional Chinese medicine which has a long application history is increasingly used and widely accepted at present [1,2]. Multiple studies have shown that they can be used for comprehensive and effective prevention, and also for treatment of many major diseases such as cardiovascular disease, diabetes mellitus, cancer, etc. [3,4,5]. Although they have good curative effects, many problems related to medicinal quality were noted, hindering their further application [6]. For example, in some traditional Chinese medicines multi-origin phenomena always exist, meaning that they originate from more than one species and share the same Chinese name [7]. In order to overcome these obstacles, researchers have paid much attention on the multi-aspect and multi-level characterization of traditional Chinese medicines based on diverse techniques and methods [8,9,10,11].

Medicinal *Coptis* plants (Ranunculaceae) have been widely used for preventing and treating human diseases for centuries in China. Their dried rhizomes, named Rhizoma Coptidis (Huanglian in Chinese) can be utilized as anti-inflammatory, antibacterial, and antidiabetic agents. [12,13]. Currently, Rhizoma Coptidis are mainly derived from several *Coptis* plants including *Coptis chinensis* Franch., *C. deltoidea* C.Y. Cheng et Hsiao., *C. omeiensis* C.Y. Cheng or *C. teeta* Wall., which are mostly distributed in Southwest China and the wild resources are almost endangered [14]. Comparatively, *C. chinensis* is a common raw material with a higher cultivated production. The other three species are always used as the substitutes because of their similar medicinal constituents and therapeutic effects, however, in order to perform a better-quality assessment, it is essential to develop a reliable strategy for characterizing the variation among these *Coptis* plants for a reasonable exploitation of these medicinal resources.

Nowadays, secondary metabolites have been demonstrated to product a variety of bioactivities for protecting human health. Phytochemical and pharmacological studies on *Coptis* plants have indicated that their metabolites are very similar and the major active components are protoberberine alkaloids including berberine, palmatine, jatrorrhizine, coptisine, columbamine and epiberberine [13,15]. These constituents display many medicinal properties such as antiviral, anti-inflammatory and antimicrobial activities [16,17,18]. In recent years, many analytical methods including high performance liquid chromatography-ultraviolet (HPLC-UV), ultra-performance liquid chromatography (UPLC) and proton nuclear magnetic resonance (^1^H-NMR) have been used to analyze these compounds in *Coptis* plants [19,20,21]. These studies have shown that their contents are obviously different among different species. For instance, the content of berberine which is regarded as a marker for quality control, is relatively higher in *C. teeta* and *C. chinensis* according to the research by He et al. [14]. However, these metabolic data from previous studies was not deeply interpreted, and multivariate statistical methods which contribute to search for the features of metabolic data should be appropriately applied for the characterization of main active metabolites in *Coptis* plants.

In addition to metabolic strategies, molecular genetics methods can perhaps further elucidate genetic divergence among different species, which is the foundation of phytochemical differentiation [22]. With the advent of next-generation sequencing (NGS) technologies, they have been proven useful for high-throughput genetic marker production and the large-scale discovery of genome-wide Single Nucleotide Polymorphisms (SNPs) in complex genomes [23]. SNPs are considered to be the most reliable genetic markers, with advantages of flexibility, rapidity and low error rate. Restriction site-associated DNA sequencing (RAD-seq) is one of the most recently developed genotyping methods based on NGS. It is a time and cost-effective technique which can generate sequence data adjacent to a large number of restriction enzyme digestion sites and reduce the complexity of large genomes for an easier SNP discovery. Moreover, this technique has been applied to identify genetic variants among different species with or without a published reference genome [24,25]. For example, different *Citrus* species have been analyzed by this method [26], which is able to discriminate citrus varieties from closely related species. In addition, it has been proven to provide SNP data for other aims such as population genetics [27], species identification [28] and genetic mapping [29]. To the best of our knowledge, no study has been conducted for the characterization of *Coptis* plants in terms of RAD-seq technique.

As the final response to gene expression, the levels of metabolic products is regulated by genes to some extent. The information from genetic and metabolic platforms may have some relevance. In present, the combination of multi-omics techniques were commonly used to interpret the molecular and chemical characterizations of medicinal plants. In view of these aspects, the combination of these approaches can provide a more powerful evidence to search for the relationships among *Coptis* plants with the aid of multivariate statistical methods. In this study, we applied two low-cost and time effective techniques (HPLC-UV and RAD-seq) for the identification and characterization of four *Coptis* plants (*C. chinensis*, *C. deltoidea*, *C. omeiensis* and *C. teeta*) based on metabolic and molecular methods. HPLC-UV was used for quantitative determination of eight key alkaloids in these plants. Meanwhile, RAD-seq was applied to generate SNP data to characterize the difference in genome with respect to four species. Applying this comprehensive strategy, the objective of this study was going to develop a reliable and effective approach for the characterization of *Coptis* medicines to better control their quality.

## 2. Results

### 2.1. Results of Validation for HPLC-UV Method

The results of calibration curves, *r*^2^, LOD and LOQ are shown in Table 1. For all alkaloids, the correlation coefficient values (*r*^2^ ≥ 0.999) indicate good linear regressions within the investigated ranges. The relative standard deviations (RSD) are in the range of 0.29–1.95%, indicating an excellent stability and repeatability of this method. Recovery test was carried out to evaluate the accuracy of this method. The result for alkaloid compounds are ranged from 97.52% to 103.35% with RSDs less than 2.16%. According to the results of precision, stability, repeatability and recovery test (Appendix A), the method was appropriate for the simultaneous detection and quantification of the eight alkaloids in *Coptis* rhizomes.

### 2.2. HPLC-UV Determination of Eight Alkaloids

As shown in Figure 1, the main eight peaks are evaluated and identified as magnoflorine, groenlandicine, jatrorrhizine, columbamine, epiberberine, coptisine, palmatine and berberine by the comparison with reference standards, respectively. The quantitative accumulations of these eight compounds are presented in Table 2. The result indicates that eight alkaloids can be found in all samples with the exception of epiberberine and their contents are obviously varied among different species. In the light of previous reports, berberine, coptisine, jatrorrhizine and palmatine constitute the main protoberberine alkaloids of *C. chinensis*, *C. deltoidea*, *C. omeiensis* and *C. teeta* [30,31,32]. Berberine is the most abundant alkaloid and its concentration is ranged from 48.66 ± 3.24 to 93.46 ± 5.12 mg/g. The extensive researches on this compound have demonstrated that it has many pharmacological properties and multi-aspect therapeutic effects such as antimicrobial, antiprotozoal, anticancer, antidiabetic, anti-inflammatory and cardiovascular activities [33]. With respect to other alkaloids, coptisine (12.02 ± 1.26–19.84 ± 2.40 mg/g), palmatine (6.21 ± 0.53–15.39 ± 1.92 mg/g) and jatrorrhizine (3.88 ± 0.55–10.06 ± 1.09 mg/g) are also the highly existed compounds. 

In the rhizomes derived from *C. omeiensis* and *C. teeta*, epiberberine is not detected. Rhizome of *C. teeta* has the highest concentration of berberine (93.46 ± 5.12 mg/g). Besides, the accumulations of jatrorrhizine (10.06 ± 1.09 mg/g) and magnoflorine (7.33 ± 0.67 mg/g) are significantly higher than that from other three plants. With respect to *C. chinensis*, it is the best source for epiberberine (11.21 ± 1.75 mg/g), coptisine (19.84 ± 2.40 mg/g), palmatine (15.39 ± 1.92 mg/g) and columbamine (3.79 ± 0.92 mg/g). Notably, epiberberine is the most typical component of *C. chinensis*, and its content is significantly higher than that from other *Coptis* plants. For the rhizomes separated from *C. deltoidea*, the level of groenlandicine (9.11 ± 2.36 mg/g) is markedly higher, and the lowest level of berberine (48.66 ± 3.24 mg/g) is accumulated compared to the rhizomes from other three *Coptis* plants. In the recent studies related to different *Coptis* plants [14,34], the accumulated trends of these active components are consistent with our results although there are some small differences with respect to certain compounds. The content of berberine for *C. chinensis* in recent studies is slightly higher than that in this study, and berberine in *C. teeta* is less than that in our research. These differences may be caused by the different collected origins of plant materials.

Based on the distinct difference of quantitative determination in relation to the alkaloid compositions, principal component analysis (PCA) was carried out to visualize the differences regarding the rhizomes of different plants and further explore the relation between each compound and sample clustering, which was never discussed in the previous papers. The PCA result (Figure 2) exhibits an excellent separation tendency of samples related to the first two principal components (PCs) (79.6% of the total variance (Appendix A)). As shown in Figure 2, all samples are sorted into four groups. The sample clusters of *C. teeta* and *C. deltoidea* are located in negative PC 1 scores and PC 2 makes a major contribution to differentiate them. Most samples originated from *C. omeiensis* and *C. chinensis* are assigned into positive scores of PC 1 which can successfully separate these samples. The compounds of palmatine, epiberberine, columbamine, jatrorrhizine and coptisine are the dominant variables on PC 1, thereby causing greater variability among these samples. Magnoflorine, groenlandicine, coptisine and berberine are the dominant variables on PC 2 (Figure 3a).

As shown in Figure 3b, these metabolites play a different role for differentiating rhizomes originated from different plants. Comparatively, palmatine, epiberberine and columbamine are effective for distinguishing *C. chinensis* samples from others. Magnoflorine and berberine make a great contribution for clustering samples originated from *C. teeta*. The excellent separation of rhizome from *C. deltoidea* is closely related to the level of groenlandicine. 

Referring HPLC-UV data, medicine from *C. chinensis* accumulates the highest level of palmatine, epiberberine and columbamine. Magnoflorine and berberine are mostly synthesized in samples from *C. teeta*, while *C. deltoidea* samples possess the highest accumulation of groenlandicine. Hence, these compounds can be used as the important index to differentiate corresponding *Coptis* medicines from others. To further elucidate the relationship among four *Coptis* species based on metabolic information clearly, HCA result was presented as a dendrogram plot (Figure 4). Four clear clusters are observed: cluster I corresponded to samples from *C. omeiensis*; cluster II to samples from *C. deltoidea*; cluster III to samples from *C. teeta*; and cluster IV to samples from *C. chinensis*, validating the observation in the PCA plot. In addition, there is a shorter distance between rhizomes originated from *C. omeiensis* and these from *C. deltoidea*. Then *C. teeta* samples cluster together with them, indicating that the metabolic characteristics of *C. omeiensis* samples are more similar to *C. deltoidea* than others and *C. chinensis* samples are more relatively different from other species.

### 2.3. RAD-seq Data Analysis and SNP Discovery

With the respect to the metabolic composition analysis, the difference among samples originated from four species was obvious. To characterize these samples at the molecular level, their DNA were isolated from 10 mixed samples of each species to construct RAD libraries and then successfully sequenced by NGS platform.

A total of 44,747,016 reads are generated and 43,871,958 reads are retained after discarding reads with low-quality, containing unidentified nucleotides and adapters. Initial processing result for each plant is shown in Table 3 and Appendix A. Due to the lack of reference genome information for *Coptis* plants, contigs were created by de novo assembly. The number of obtained contigs is 965,140 and the lengths for the assembly are ranged from 162 to 135,561 bp (Appendix A) with the N_50_ of 440 bp and an average length of 333 bp (Table 4). The GC dinucleotide content is 38.03% which is close to results from other paired-end RAD-seq studies in plant genomes [35,36].

The paired-end sequences of each sample were aligned to the assembled reference sequence and mapped reads were used to identify SNPs. Based on the unique characteristics of RAD-seq protocol, SNPs around restriction enzyme tag fragments were discovered, which may be related to the variations of metabolites among different species. The result indicates that a total of 2,443,407 SNPs are identified from RAD-sequence data among four *Coptis* plants. Of all the identified SNPs, 67.83% are identified as transitions (A/G or C/T) and 32.17% are classified as transversions (A/C, A/T, G/T, or C/G) (Figure 5). In general, transitions occurred more frequently than transversions due to the interchange between purine and pyrimidine nucleotide bases [37]. In addition, the observed SNP transition/transversion rate in this study is 2.11, which is similar to that reported in rice (2.3) [38].

### 2.4. Genetic Relationship among Four Coptis Plants

To reveal the genetic relationships among four *Coptis* plants, PCA and HCA were also successively performed using SNP data. The PCA result exhibits that the first two PCs account for 68.63% of the total variance. PCA plot (Figure 6a) illuminates that *C. deltoidea* and *C. omeiensis* have a closer genetic relationship. Especially, based on the PC 1, the distance between *C. deltoidea* and *C. omeiensis* is relatively close while *C. chinensis* is more adjacent with *C. teeta*. The conclusion is consistent with the HCA result from the neighbor-joining phylogenetic analysis (Figure 6b).

Four *Coptis* plants are clearly classified into two clades (Figure 6b): one clade containing *C. chinensis* and *C. teeta* while the other clade containing *C. deltoidea* and *C. omeiensis*. The result mostly agreed with the metabolic conclusion and previous studies reporting that *C. deltoidea* and *C. omeiensis* had the closest genetic distance among *Coptis* plants [12].

## 3. Materials and Methods

### 3.1. Chemicals and Reagents

HPLC grade acetonitrile was purchased from Thermo Fisher Scientific Co, Ltd. (Shanghai, China). Deionized water used for chromatographic analysis was purified with a Milli-Qultra-pure water system (Millipore, Burlington, MA, USA). Reference standards of palmatine, coptisine, epiberberine, columbamine, jatrorrhizine, magnoflorine and groenlandicine were purchased from Chroma-Biotechnology Co., Ltd. (Chengdu, China), and berberine was provided by Chinese National Institute for Food and Drug Control (Beijing, China). The purities for all compounds are higher than 98%. CTAB used for DNA extraction was purchased from Guangzhou chemical reagent factory (Guangzhou, China). *EcoR* I was obtained from New England Biolabs (Ipswich, MA, USA). All other reagents were analytical grade from Chron Chemicals Co, Ltd. (Chengdu, China).

### 3.2. Plant Material and Sample Preparation

A sample collection of 40 specimens including four *Coptis* plants (*C. chinensis*; *C. deltoidea*; *C. omeiensis* and *C. teeta*) was processed, and 10 samples of each species were prepared (Appendix A). These 5-year plant materials were collected from wild habitats with the altitude over 2000 m and identified by Professor Yuntong Ma (College of Pharmacy, Chengdu University of Traditional Chinese Medicine, Chengdu, China). Young leaf tissues of each individual plant were firstly separated and frozen in liquid nitrogen for DNA extraction. Then, the rhizome (Appendix A) was carefully washed and dried at 60 °C to constant weight. Finally, these samples were ground into powder for the metabolite analysis.

### 3.3. HPLC-UV Analysis

Dried powder samples (0.02 g) were accurately weighed and extracted with hydrochloric acid-methanol solution (10 mL, 1:100, *v/v*) for 30 min in an ultrasonic bath at room temperature. Then, the decrease in weight was replenished with the extraction solvent. The sample solution was filtered through a 0.45 μm membrane filter prior to the HPLC-UV analysis. A Shimadzu HPLC system (Shimadzu, Kyoto, Japan) equipped with a LC-20AT quaternary pump, a SIL-20A XR autosampler, a CTO-20AC column oven and a SPD-20A UV/Vis detector was used to determine the objective compounds. The chromatographic column Xtimate C18 (250 × 4.6 mm, 5 μm, Welch, Shanghai, China) was applied to perform chromatographic separation and the column temperature was constantly kept at 25 °C The binary mobile phase consisted of acetonitrile (A) and 30 mmol/L ammonium bicarbonate solution containing 0.7% ammonia and 0.25% triethylamine (B) at a continuous flow rate of 1 mL/min in the following gradient program: 0–15 min, 10%–25% A; 15–25 min, 25%–30% A; 25–50 min, 30%–45% A. The injection volume was 5 μL and the detection wavelength was acquired at 270 nm. The contents of eight alkaloids (berberine, palmatine, coptisine, epiberberine, columbamine, jatrorrhizine, magnoflorine and groenlandicine) in each sample were calculated from standard curves.

### 3.4. Validation of HPLC–UV Method

The validation of HPLC-UV method in terms of linearity, precision, repeatability, stability and recovery test is an important process for quantitative determination of major active components. The calibration curves of eight alkaloids were established by plotting the peak areas against the standard concentrations, respectively. The precision of the HPLC-UV method was investigated by five replicate measurements of same mixed standard solution. For stability test, the same solution was analyzed at the time intervals of 0, 5, 10, 15 and 20 h, respectively. In order to evaluate the repeatability, five different sample solutions independently prepared from a same sample were analyzed. The limits of detection (LOD) and limits of quantitation (LOQ) were determined at a signal-to-noise ratio (S/N) of about 3 and 10, respectively.

### 3.5. DNA Isolation, RAD Library Construction and Sequencing

Genomic DNA was isolated from frozen leaf tissues of 10 mixed samples regarding to each *Coptis* plants, using a modified CTAB method [38]. The quality and quantity of DNA were detected by 1% agarose gel electrophoresis and NanoDrop 2000 (Thermo Scientific, Waltham, MA, USA). The genomic RAD libraries were constructed using a protocol adapted from Baird et al. [39]. Briefly, each genomic DNA pooled from four *Coptis* species was digested with the restriction endonuclease *EcoR* I and then the reaction was stopped by inactivating at 65 °C Each P1 adapters with a unique 8 bp multiplex identifier sequence (MID) was ligated to the products of restriction reaction. After ligations of P1 adaptors, the DNA samples containing the different MIDs were pooled in proportionate amounts. Then, the products were randomly sheared to DNA fragments. Subsequently, adaptor-ligated DNA amplicons were purified and electrophoresed through 1.5% agarose gel, and the DNA fragments in the range 300–500 bp were isolated. After end repair, purification and elution, dATP overhangs were added to the DNA fraction. A paired-end P2 adapter containing T overhangs was ligated to the P1-ligated DNA template with a specific adapter. Only the fragments with both P1 and P2 adapters could be successfully amplified. The ligated material was then subjected to PCR enrichment. And then, the constructed RAD libraries were sequenced by Illumina HiSeq4000 sequencing platform (Illumina, San Diego, CA, USA) and 150 bp paired-end reads were generated. Finally, the sequencing data for each sample was extracted according to the specific MID.

### 3.6. De Novo Assembly of RAD Tags and SNP Identification

All raw reads were checked for the presence of expected *EcoR* I motif (AATTC) following the sample-specific MID. To obtain clean and high quality reads, three stringent filtering standards (removing reads aligned to the barcode adapter, with more than 10% unidentified nucleotides and contained >50% low-quality bases (phred quality scores of ≤20)) were conducted. Retained first and second high quality reads were used for next downstream analysis.

Because of the unavailable reference of *Coptis* genome sequence, the identification of SNPs was implemented *de novo* [40,41]. Briefly, the primary steps were roughly as follows:(1)After removing the MID and the restriction site, the information of individual stacks was built by sorting and clustering similar first reads from each sample.(2)The individual stacks were clustered and the consensus sequences were obtained.(3)According to the clusters mentioned above, the second reads could be sorted into groups and separately assembled to contigs, which were joined with the first reads consensus sequences. The paired-end reads were used to construct scaffolds serving as a reference for the following analysis.

To identify SNPs, the clean reads from the paired-end sequences of each sample were aligned to the assembled reference sequence using Burrows-Wheeler Aligner (BWA) (version 0.7.16a–r1181; http://bio-bwa.sourceforge.net/bwa.shtml) [40]. Then the aligned reads were converted to Binary Alignment/Map (BAM) files using SAMtools (http://samtools.sourceforge.net/), and Picard program (version 1.129; http://broadinstitute.github.io/picard/) was used to sort, index and remove duplicates. SNPs were called out for each *Coptis* plants against the reference sequence using the GATK’s Unified Genotyper (version 3.4-46, https://software.broadinstitute.org/gatk/). And then, SNPs were further filtered using GATK’s Variant Filtration with proper standards.

### 3.7. Statistical Analysis

A one-way analysis of variance based on a Tukey’s multiple comparison test was firstly applied to determine the content variations of alkaloid compounds among the rhizomes from four *Coptis* plants. The statistical significance was determined on a 95% probability level (*p* < 0.05).

PCA is a non-parametric and unsupervised multivariate technique for reducing the dimensionality of original datasets, increasing interpretability but minimizing information loss at the same time [42]. It can do so by creating several new uncorrelated variables called PCs [43]. Generally, the relevant information from the complex data can be mostly obtained using the first two PCs. Hence, a two-dimensional projection was constructed to obtain an overview of samples from four plants. Meanwhile, in order to further explore the data characteristics, a loading plot was developed to visualize the relevance between each PC and studied objectives.

HCA is also an exploratory technique where the groups are sequentially created by systematically merging similar clusters together. According to metabolic or molecular data, the property of samples in the same group is similar whereas the samples in different groups are different. Because it only refers a distance parameter (Mahalanobis distance in our study), it is more visualized to elucidate relationship among different samples by clustering samples into a one-dimensional dendrogram. Currently, this method has been extensively used to evaluate the multivariate association between bioactive compounds and samples [44]. In this study, HCA was applied to validate PCA result and further elucidate the inter-species relationship among different *Coptis* plants according to the metabolic data. Additionally, this method was also used to calculate the genetic distances among different species based on SNPs from the RAD data, and a phylogenetic tree was constructed using a neighbor-joining method in terms of the generated distance matrix, with bootstrap values at the default setting of 100 trials. The data analysis flow chart of this study is shown in Appendix A.

## 4. Conclusions

In this study, HPLC-UV and RAD-seq were applied together for a metabolic and molecular characterization of four *Coptis* medicinal plants. On the basis of metabolic profiles, the accumulation of main alkaloid compounds among rhizomes originated from different *Coptis* plants obviously varies and berberine is the most abundant alkaloid compound in all samples. PCA and HCA were used to deeply explore data characters and successfully visualize the metabolic variation and relationship among rhizomes from four *Coptis* plants. In addition, we have demonstrated paired-end RAD-seq is an efficient approach for SNP discovery. From Illumina paired-end sequence data, over 90,000 high-quality sequence contigs with an N_50_ length of 440 bp were assembled and 2,443,407 SNPs from four *Coptis* plants were identified. Based on SNP data, genetic relationship was also illuminated that *C. deltoidea* and *C. omeiensis* had the closest genetic relationship. In summary, results from this study can provide a comprehensive metabolic and molecular genetic profiling for the characterization of different *Coptis* plants. Furthermore, it will contribute to more stringent quality control and reasonable exploitation of these resources.

## Figures and Tables

**Figure 1 molecules-23-03090-f001:**
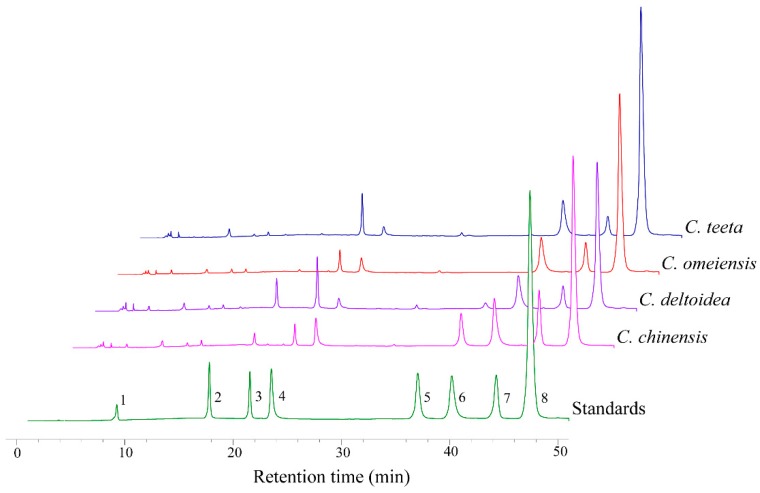
HPLC-UV chromatogram of eight alkaloid compounds in rhizomes separated from different *Coptis* species (Peaks 1–8 indicate magnoflorine, groenlandicine, jatrorrhizine, columbamine, epiberberine, coptisine, palmatine and berberine, respectively).

**Figure 2 molecules-23-03090-f002:**
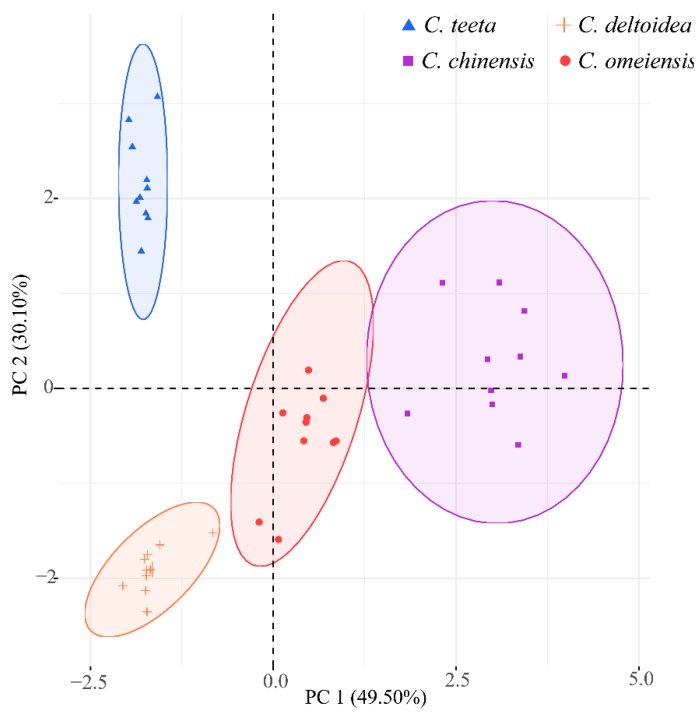
PCA score plot of different *Coptis* medicines based on quantitative determination of alkaloid compounds.

**Figure 3 molecules-23-03090-f003:**
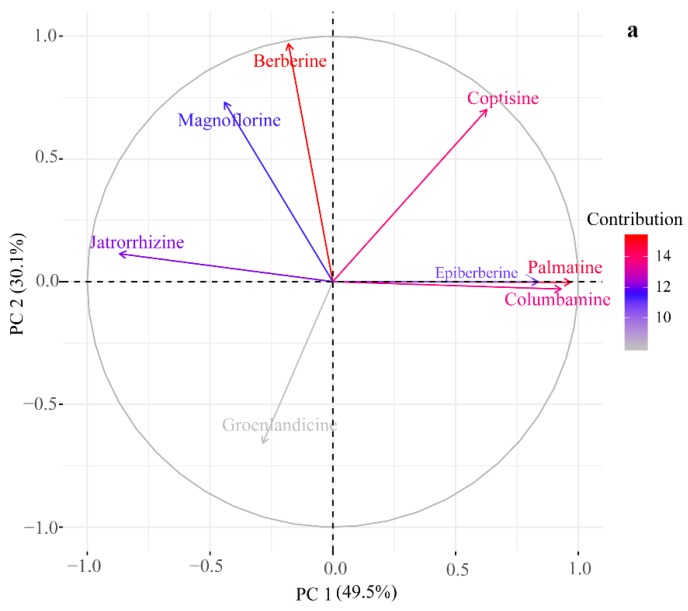
(**a**) Loading plot of PCA; (**b**) Score of each variable on PC 1 and PC 2.

**Figure 4 molecules-23-03090-f004:**
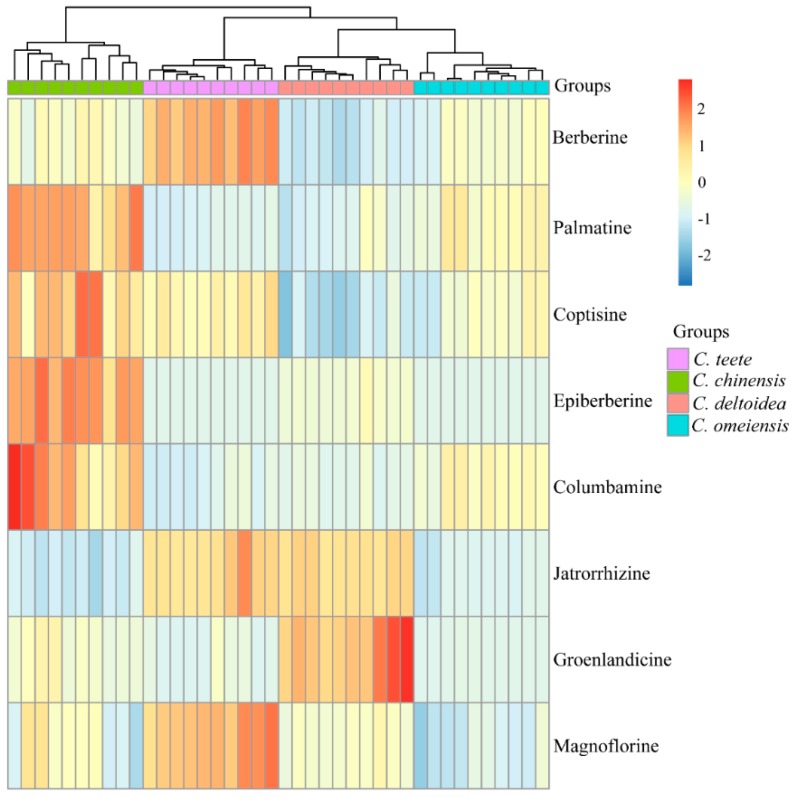
Result of HCA according to the concentration of eight alkaloids.

**Figure 5 molecules-23-03090-f005:**
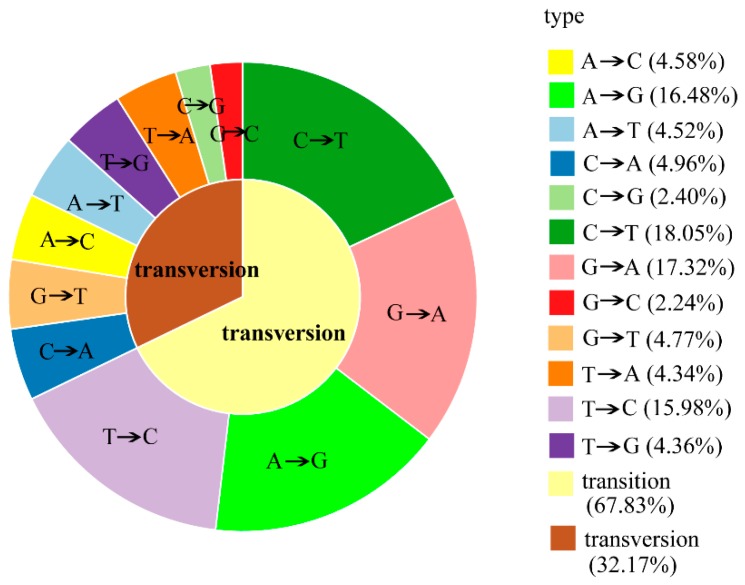
Ratio of SNP transitions and transversions observed in *Coptis* plants.

**Figure 6 molecules-23-03090-f006:**
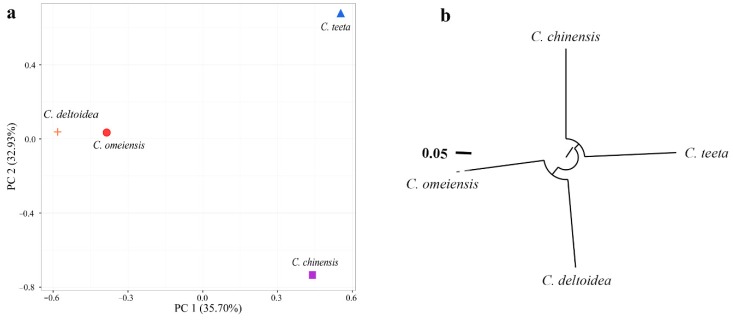
(**a**) Principal component analysis of the four *Coptis* plants based on identified SNPs; (**b**) Neighbor-joining tree based on identified SNPs.

**Table 1 molecules-23-03090-t001:** Levels of alkaloid compounds in rhizomes originated from four *Coptis* plants.

Standard	Calibration Curve	Linearity (μg/mL)	*r* ^2^	LOD (μg/mL)	LOQ (μg/mL)
Berberine	y = 18906563.4500 x − 18038.6417	55.29–470.00	0.9998	7.24	24.13
Palmatine	y = 19332328.3900 x + 6058.2880	4.47–54.32	0.9998	0.69	2.32
Jatrorrhizine	y = 14849036.3600 x + 241.1122	2.95–35.84	0.9998	0.51	1.69
Coptisine	y = 19176427.8100 x − 2518.8334	5.69–69.16	0.9999	0.61	2.04
Columbamine	y = 31118373.4300 x − 2025.8588	2.42–29.40	0.9998	0.39	1.31
Epiberberine	y = 19026602.7700 x + 4470.7873	5.01–86.81	0.9999	1.11	3.70
Magnoflorine	y = 4769076.5410 x + 293.8634	3.25–56.40	0.9998	0.82	2.73
Groenlandicine	y = 13999875.3500 x − 6591.2541	2.63–65.20	0.9999	0.66	2.19

**Table 2 molecules-23-03090-t002:** Contents of alkaloid compounds (mg/g) in rhizomes originated from four *Coptis* plants.

Alkaloid	Species			
	*C. omeiensis*	*C. teeta*	*C. chinensis*	*C. deltoidea*
**Ber**	62.17 ± 5.36a	93.46 ± 5.12b	64.93 ± 4.97a	48.66 ± 3.24c
**Pal**	9.94 ± 1.65a	6.21 ± 0.53b	15.39 ± 1.92c	6.72 ± 1.45b
**Jat**	4.51 ± 0.56a	10.06 ± 1.09b	3.88 ± 0.55a	9.84 ± 0.44b
**Cop**	15.07 ± 1.65a	17.23 ± 1.07b	19.84 ± 2.40c	12.02 ± 1.26d
**Col**	2.37 ± 0.33a	1.44 ± 0.26b	3.79 ± 0.92c	1.64 ± 0.16b
**Epi**	N	N	11.21 ± 1.75a	1.95 ± 0.74b
**Mag**	3.17 ± 0.63a	7.33 ± 0.67b	4.40 ± 1.26c	4.57 ± 0.35c
**Gro**	0.74 ± 0.09a	0.95 ± 0.80a	2.66 ± 1.11b	9.11 ± 2.36c

Different superscript letters within the same row or column indicate significant differences at *p* < 0.05 according to a Tukey’s multiple comparison test. Undetected compound is marked as N. (Ber: berberine; Pal: palmatine; Jat: jatrorrhizine; Cop: coptisine; Col: columbamine; Epi: epiberberine; Mag: magnoflorine; Gro: groenlandicine).

**Table 3 molecules-23-03090-t003:** Number of reads retained after initial quality filtering process for each *Coptis* plants sequenced.

Species	Raw Reads Number	Clean Reads Number (%)	Read Length	Adapter (%)	Low Quality (%)
*C. omeiensis*	10,872,358	10,655,424 (98%)	150/150	114,860 (1.06%)	98,434 (0.91%)
*C. teeta*	11,100,218	10,874,838 (97.97%)	150/150	125,744 (1.13%)	95,970 (0.86%)
*C. chinensis*	11,795,008	11,548,500 (97.91%)	150/150	110,340 (0.94%)	132,294 (1.12%)
*C. deltoidea*	10,979,432	10,793,194 (98.3%)	150/150	64,318 (0.59%)	118,454 (1.08%)
Total	44,747,016	43,871,958	-	415,262	445,152

**Table 4 molecules-23-03090-t004:** Result of paired-end RAD-seq assembly statistics.

Feature	Value
Number of assembled reference sequence contigs	965,140
Total assembly length (bp)	321,905,653
Minimum contig length (bp)	162
Maximum contig length (bp)	135,561
GC%	38.03
N_50_ Contig Length (bp)	440
N_90_ Contig Length (bp)	170

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
