# Peer review of "A Multi-Level Strategy Based on Metabolic and Molecular Genetic Approaches for the Characterization of Different Coptis Medicines Using HPLC-UV and RAD-seq Techniques"

_molecules, 2018, doi:10.3390/molecules23123090_

Round 1

Reviewer 1 Report

This paper describes a multi-level strategy based on metabolic and molecular genetic approaches for the characterization of four Coptis species.Coptis species are very important medicinal herbs for Chinese medicine. For manufacturing the medicine, it is important to have the reliable strategy to characterize closely related Coptis species for a better assessment. The authors succeeded in developing the reliable and effective methods for the characterization of four different Coptis species by combining HPLC-UV and RAD-seq techniques. The methods described in this paper is useful and should be shared by researchers in this filed. The experiments are carefully performed and the conclusions are reasonable. As a conclusion, the paper is sufficient to merit publication in “molecules” though a minor revision is recommended which needs to include the following points. 

(1) Move “2.1 Validation of HPLC-UV methods” to materials and methods section. 

(2) Figure2 and Figure 6: Please use same symbol and color for each species in both figures. 

(3) Please make the text size of the figures bigger.

(4) Figure S3: It is too small to see. Need more explanation.  The Figure S3 may not be needed in the paper. 

(5) Line 278 - 283: Font size is different. 

(6) The NJ tree (figure 6b) should be out of the PCA graph. 

(7) Please mention if the conclusions obtained from tow methods (HPLC-UV and RAD-seq) are consistent or not. 

(8) The author performed two different methods to characterize the 4 species and claimed that it is multi-level strategy. I agree with it though I think it is better to mention the advantages or merit to combine the two methods. 

Author Response

Dear Editor,

Thank you for your and reviewers’ comments concerning our manuscript entitled “A multi-level strategy based on metabolic and molecular genetic approaches for the characterization of different Coptis medicines using HPLC-UV and RAD-seq techniques”. We have carefully revised the manuscript according to your and reviewers’ suggestions. The point to point responds to the reviewers’ comments are listed as following:

Point 1: Move “2.1 Validation of HPLC-UV methods” to materials and methods section.  

Response 1: Thanks for your valuable advice. We have changed these structures in the revised paper.

Point 2: Figure2 and Figure 6: Please use same symbol and color for each species in both figures.

Response 2: We have modified the symbol and color for each species in Figure 6a. Thanks for your valuable advice.

Point 3: Please make the text size of the figures bigger.

Response 3: We have made the text size of all figures bigger in the revised paper. Thanks for your valuable advice.

Point 4: Figure S3: It is too small to see. Need more explanation. The Figure S3 may not be needed in the paper.

Response 4: Thanks for your valuable advice. We have removed Figure S3 from supplementary materials.

Point 5: Line 278 - 283: Font size is different.

Response 5: We are sorry for our negligence. We have corrected the font size in the revised paper.

Point 6: The NJ tree (figure 6b) should be out of the PCA graph.

Response 6: Thanks for your valuable advice. We have separately presented Figure 6a and Figure 6b in the revised paper.

Point 7: Please mention if the conclusions obtained from two methods (HPLC-UV and RAD-seq) are consistent or not.

Response 7: The conclusions obtained from two methods (HPLC-UV and RAD-seq) are consistent. In this paper, based on metabolic information from HPLC-UV and SNPs from the RAD data, we have successfully illustrated the characterization of four Coptis plants. In terms of both metabolic and molecular genetic profiling, we find that the metabolic products of C. deltoidea and C. omeiensis are similar, and the two species also have the closest genetic distance among Coptis plants. Hence, the phytochemical differentiation of four Coptis plants supported the genetic divergence of them inferred from RAD-seq data.

Point 8: The author performed two different methods to characterize the 4 species and claimed that it is multi-level strategy. I agree with it though I think it is better to mention the advantages or merit to combine the two methods.

Response 8: In this paper, HPLC-UV was used for quantitative determination of eight key alkaloids in four Coptis plants, which revealed the variation of metabolic profiles among these plants. With respect to RAD-seq, it was applied to generate SNP data to characterize the difference in genome, which illustrated the molecular characterization of four Coptis plants. Based on the study of metabolic profiles, molecular genetic methods perhaps can further elucidate genetic divergence among different species, which is the foundation of phytochemical differentiation. Therefore, the combination of two different methods can be mutual validated from both metabolic and molecular aspects. We have added these contents in the revised paper. Thank you for your valuable advice.

Reviewer 2 Report

This manuscript reports results on metabolic and molecular genetic approaches for the characterization of different Coptis medicines using HPLC-UV and RAD-seq techniques.

The present manuscript is well-written based on their data. There are some questions and/or suggestions to improve the quality of the manuscript.

1. Authors need to compare their developed HPLC-UV method with those of previous literatures and discuss them somewhere.

2. Please indicate the purity for each compound.

3. Please generate more detail information about sample preparation for HPLC-UV

4. Authors should need to discuss how to apply and use their metabolic and molecular genetic approaches for the characterization of different Coptis medicines using HPLC-UV and RAD-seq techniques more specifically with a detail somewhere.

5. Please check several errors and typos carefully

Author Response

Dear Editor,

Thank you for your and reviewers’ comments concerning our manuscript entitled “A multi-level strategy based on metabolic and molecular genetic approaches for the characterization of different Coptis medicines using HPLC-UV and RAD-seq techniques”. We have carefully revised the manuscript according to your and reviewers’ suggestions. The point to point responds to the reviewers’ comments are listed as following:

Point 1: Authors need to compare their developed HPLC-UV method with those of previous literatures and discuss them somewhere.

Response 1: In the recent studies related to different Coptis plants [1,2], the accumulated trends of these active components are consistent with our results although there are some small differences in the quantity with respect to certain compounds. The content of berberine for C. chinensis in recent studies is slightly higher than that in this study, and berberine in C. teeta is less than that in our research. The differences may be caused by the different collected origins of plant materials. We have added these contents in the revised paper. Thanks for your valuable advice.

[1] He, Y.; Hou, P.; Fan, G.; Arain, S.; Peng, C. Comprehensive analyses of molecular phylogeny and main alkaloids for Coptis (Ranunculaceae) species identification. Biochem. Syst. Ecol. 2014, 56, 88-94.

[2] Lv, X.; Li, Y.; Tang, C.; Zhang, Y.; Zhang, J.; Fan, G. Integration of HPLC-based fingerprint and quantitative analyses for differentiating botanical species and geographical growing origins of Rhizoma Coptidis. Pharm. Biol. 2016, 54, 3264-3271.

Point 2: Please indicate the purity for each compound.

Response 2: The purity for each reference standard is higher than 98%. We have added this information in the revised paper.

Point 3: Please generate more detail information about sample preparation for HPLC-UV.

Response 3: We have presented the detail information in the section of “Sample Preparation” in the revised paper.

Point 4: Authors should need to discuss how to apply and use their metabolic and molecular genetic approaches for the characterization of different Coptis medicines using HPLC-UV and RAD-seq techniques more specifically with a detail somewhere.

Response 4: As the final response to gene expression, the levels of metabolic products are regulated by genes to some extent. The information from genetic and metabolic platforms may have some relevance. The combination of these approaches can provide a more powerful evidence for revealing the relationships among these Coptis plants. Hence, we collected the information from these platform and multivariable statistics methods were applied to explore these data and illustrate their structure features. The results from these techniques and methods are also validated and compared in order to obtain a more comprehensive result.

In addition, multi-omics techniques were also jointly used to describe the molecular and chemical characterizations of medicinal plants extensively in present. Hence, we applied two effective techniques of HPLC-UV and RAD-seq for the identification and characterization of Coptis species, which is the first work for these plants. We have added some discussion in the revised paper.

Point 5: Please check several errors and typos carefully

Response 5: We have checked and corrected these mistakes in the revised paper. Thanks for your valuable advice.

Reviewer 3 Report

In this manuscript, Ma and coworkers showed the characterization of a traditional Chinese medicine Coptis from the perspective of metabolite composition and genetic variations. This is an important work from the perspective of quality control of Chinese medicine. I suggest this manuscript could be considered for publication in Molecules after revision.

1. The language of the manuscript needs to be further polished by native speakers.

2. Page 2, line 46 "can be utilized to clear away heat, dry dampness and purge the sthenic fire". These are concepts in Chinese medicine, it is better to explain them in the context of diseases that could be understood internationally.

3. The HPLC-UV data showed metabolite variations between different species of Coptis, but which composition is the most therapeutically active?

4. For the SNP data, it is straightforward to get the difference between the species. But what is the relationship between SNP and therapeutic efficacy?

Author Response

Dear Editor,

Thank you for your and reviewers’ comments concerning our manuscript entitled “A multi-level strategy based on metabolic and molecular genetic approaches for the characterization of different Coptis medicines using HPLC-UV and RAD-seq techniques”. We have carefully revised the manuscript according to your and reviewers’ suggestion. The point to point responds to the reviewer’s comments are listed as following:

Point 1: The language of the manuscript needs to be further polished by native speakers.

Response 1: This article have been carefully checked by some experienced experts. Some mistakes have been corrected in the revised paper.

Point 2: Page 2, line 46 "can be utilized to clear away heat, dry dampness and purge the sthenic fire". These are concepts in Chinese medicine, it is better to explain them in the context of diseases that could be understood internationally.

Response 2: Thanks for your valuable advice. We have made the following modification: "can be utilized as anti-inflammatory, antibacterial, antidiabetic, and etc. " in the revised paper.

Point 3: The HPLC-UV data showed metabolite variations between different species of Coptis, but which composition is the most therapeutically active?

Response 3: In terms of this paper and previous reports, berberine is the most abundant protoberberine alkaloid in Coptis plants [1]. The extensive researches on this compound in Coptis plants have demonstrated that it has many pharmacological properties and multi-aspect therapeutic effects such as antimicrobial, antiprotozoal, anticancer, antidiabetic, anti-inflammatory and cardiovascular activities [2]. We have added these contents in the revised paper.

[1] Yang, Y.; Peng, J.; Li, F.; Liu, X.; Deng, M.; Wu, H. Determination of alkaloid contents in various tissues of Coptis chinensis Franch. by reversed phase-high performance liquid chromatography and ultraviolet spectrophotometry. J. Chromatogr. Sci. 2017, 55, 556-563.

[2] Parameswara, R. V.; Subhashis, C.; Sanjay, S. Berberine: a potential phytochemical with multispectrum therapeutic activities. Expert Opin. Investig. Drugs 2010, 19, 1297-1307.

Point 4: For the SNP data, it is straightforward to get the difference between the species. But what is the relationship between SNP and therapeutic efficacy?

Response 4: SNPs are genetic markers of choice for both linkage and association mapping and for population structure and evolution analysis [1]. They are evenly distributed along the genome. Discovery of SNPs reveals differences in the genome of four Coptis plants. Secondary metabolites, the basis of therapeutic effects, are the final products of long and complex pathways, regulated by the corresponding genes [2]. Therefore, the research on genome is the basis for the study of plant secondary metabolites synthesis and their therapeutic effects.

[1] Boutet, G.; Carvalho, S. A.; Falque, M.; Peterlongo, P.; Lhuillier, E.; Bouchez, O.; Lavaud, C.; Pilet-Nayel, L.; Rivière, N.; Baranger, A. SNP discovery and genetic mapping using genotyping by sequencing of whole genome genomic DNA from a pea RIL population. BMC genomics 2016, 17(1), 121.

[2] Jehan T.; Lakhanpaul S. Single Nucleotide Polymorphism (SNP)-Methods and applications in plant genetics: A review. Indian J. Biotechnol. 2006, 5, 435-459.
